# Relationship between serum uric acid levels and periodontitis—A cross-sectional study

**Jingjing Bai[1,2], Chenying Zhou[1,2], Ye Liu[1,2], Ming Ding[2], Zhonghua Zhang[2], Zhu Chen[2,3], Ping Feng[1,2]\*, Jukun Song[4]\***

1 Department of Periodontics, Guiyang Stomatological Hospital, Guiyang, Guizhou, China, 2 School of Stomatology, Zunyi Medical University, Zunyi, Guizhou, China, 3 Department of Endodontics and Dentistry, Guiyang Stomatological Hospital, Guiyang, Guizhou, China, 4 Oral and Maxillofacial Surgery, Stomatological Hospital Affiliated to Guizhou Medical University, Guiyang, Guizhou, China

\* 398994584@qq.com (PF); songjukun@163.com (JS)

## Abstract

### Objectives

Whether there is an association between serum uric acid level (sUA) and periodontitis remains unclear. The aim of this study was to investigate the association between moderate/severe periodontitis and sUA in US adults.

### Materials and methods

A total of 3398 participants were included in the National Health and Nutrition Examination Survey (NHANES) from 2009 to 2014. The independent variable was sUA and the dependent variable was periodontitis. SUA for continuous variables, periodontitis as classification variables. Covariate including social demographic variables, life style, systemic diseases, etc. Multiple linear regression models were used to investigate the distribution of differences in covariates between different independent groups. To investigate the association between serum uric acid levels and moderate/severe periodontitis, three models were used (Model 1: unadjusted model; Model 2: adjusted for age, sex, and race/ethnicity; Model 3: adjusted for age, sex, race/ethnicity, education, household income/poverty ratio, smoking behavior, alcohol consumption, dental floss frequency, obesity, hypertension, diabetes, high cholesterol, hyperlipidemia, and sleep disorders).

### Results

Among the 3398 patients, 42.5% had moderate/severe periodontitis. Multivariate logistic regression analysis showed that sUA was significantly associated with moderate/severe periodontitis (OR = 1.10, 95%CI: (1.03, 1.16), P = 0.0020) after adjusting for potential confounding factors. In addition, it may vary by race/ethnicity and gender. The association between sUA levels and the prevalence of periodontitis was U-shaped in women and non-Hispanic blacks.

**Data Availability Statement:** All relevant data are within the paper and its Supporting information files.

**Funding:** The author(s) received no specific funding for this work.

**Competing interests:** The authors have declared that no competing interests exist.

## Conclusion

sUA level is associated with moderate to severe periodontitis. However, the association between sUA levels and the occurrence of periodontitis in women and non-Hispanic blacks followed a U-shaped curve.

## Clinical relevance

sUA may directly or indirectly contribute to the global burden of periodontal disease, but there is little evidence that sUA is directly related to periodontitis.This study further supports that high uric acid levels are closely related to periodontitis and may contribute to the control of periodontitis. It also provides new insights into whether it can be used as an indicator to assess the risk or progression of periodontitis. More studies are needed to confirm the relationship between sUA and periodontitis.

## Introduction

Periodontitis is a chronic multifactorial inflammatory disease characterized by progressive destruction of dental supporting tissues, eventually leading to tooth loss [1]. Patients with severe periodontitis account for 23.6% of the global population, which is a highly prevalent disease [2]. Periodontitis not only affects oral health, but also has a two-way relationship with many chronic diseases, including diabetes, hypertension, Alzheimer's disease and coronary heart disease [3–6]. These chronic conditions impose a substantial health and economic burden. Plaque biofilm is a pathogenic factor of periodontitis, but the host immune response determines the severity of its destruction [7]. The host immune system is regulated by genetic, environmental, and behavioral factors [8], and impaired immune and metabolic responses induced by periodontal pathogens are key features in the pathogenesis of periodontitis [7]. Systemic oxidative stress is an important mechanism for the progression of periodontitis [9].

Uric acid (UA), the end product of purine nucleoside catabolism, is considered to be an important free radical scavenger in the antioxidant pool [10, 11]. UA is mostly present in the form of urate, which activates the crystallization of uric acid in the joint and extra-articular regions [10]. Hyperuricemia occurs when UA production exceeds excretion [12]. Studies have shown that hyperuricemia is a risk factor for a variety of inflammatory diseases such as diabetes, metabolic syndrome and cardiovascular disease [12–14]. Elevated UA levels within the normal range are considered to be a protective factor of bone mineral density against bone resorption, and exceeding normal levels of UA is a risk factor for osteoporosis and fractures, which is related to the imbalance between anti-oxidation and oxidative stress [15]. Some studies have shown that elevated levels of uric acid will change the purine metabolism of oral diseases, including absorption of periodontitis and teeth [16, 17].

It has been found that uric acid may play a role in the inflammatory pathology of periodontitis [17–19]. The level of UA in blood and saliva decreased after periodontal treatment [17, 20]. Intervention using urate-lowering drugs has also shown beneficial effects in periodontitis animals [18]. These findings tend to favor some association between elevated serum uric acid levels and periodontitis. Serum uric acid level may aggravate periodontal tissue destruction by enhancing proinflammatory cytokines [21], and oxidative stress [18]. Another part of scholars hold the opposite opinion, that is, high uric acid level has a protective effect on bone metabolism disorders and is independent of physiological bone metabolism [22].However, the impact

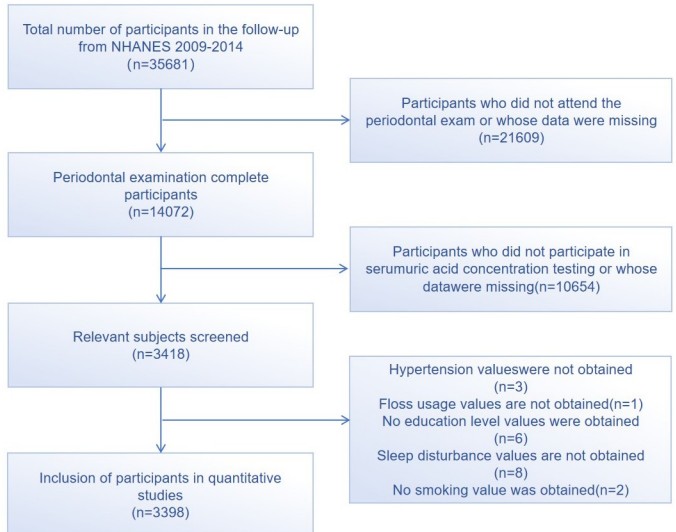

**Fig 1. Sample screening flow chart.**

of sUA on periodontitis has not been discussed as extensively in the medical literature as that of blood glucose, and we aimed to investigate the association between sUA and moderate/severe periodontitis using data from the National Health and Nutrition Examination Survey (NHANES) dataset (2009–2014). In our study, understanding the relationship between periodontitis and sUA may contribute to the prevention and control of periodontitis.

## Materials and methods

This was a cross-sectional study using the published data set from NHANES. NHANES is a stratified, multistage, nationally representative, continuous sample survey of civilian, non-institutionalized populations in fifty states and the District of Columbia [23]. NHANES data have been provided biennium since 1999. Each year, the survey assesses a nationally representative sample of approximately 5,000 people. Ethical approval and review were not sought or obtained for this study given that it was secondary analysis of publicly available, de-identified data.

### Study population

We used the 2009–2014 survey cycle of NHANES. Details can be found elsewhere [23]. The protocol required informed consent from all participants, was approved by the ethics review board at the National Center for Health Statistics, and the data were publicly available. About NHANES laboratory/medical technical personnel method, anthropometry, laboratory procedures, a blood sample collection, storage, diagnosis, and quality standard of detailed information, see www.cdc.gov/nchs/nhanes.htm. 32,283 participants who did not have complete measurement data or whose health status did not meet the inclusion criteria were excluded. Finally, 3398 participants with completed data were analyzed in this study (Fig 1).

### Periodontal examination and classification

In the data set (NHANES 2009–2014), all participants were examined by a trained and calibrated dentist. Both probing pocket depth (PD) and clinical attachment loss (CAL) showed

good reliability. The periodontist probed each tooth at six locations (mesial buccal, distal buccal, buccal, mesial lingual (or palate), distal lingual (or palate), and midlingual (or palate) during the periodontal examination. Here, we excluded third molars and examined only 28 teeth and 168 sites per person.

We referred to the CDC/AAP(Centers for Disease Control and Prevention and American Academy of Periodontology) case definition of periodontitis [24].

In order to reduce the bias caused by the high prevalence of mild periodontitis in the population, we divided patients with moderate and severe periodontitis into periodontitis categories. Patients with mild or no periodontitis were classified in the referent category [25].

Severe/moderate periodontitis was defined as follows [26]: Moderate periodontitis was defined as 2 interproximal sites with AL $\geq$ 4 mm (not on the same tooth), or $\geq$ 2 interproximal sites with PD $\geq$ 5 mm (not on the same tooth). Severe periodontitis was defined as 2 interproximal sites with AL $\geq$ 6 mm (not on the same tooth) and $\geq$ 1 interproximal site with PD $\geq$ 5 mm.

## Uric acid concentration

Uric acid DxC800 was used to measure the concentration of serum uric acid by timing endpoint method [27]. Uric acid is oxidized by uricase to produce uric acid and hydrogen peroxide. Hydrogen peroxide reacts with 4-aminoantipyrine (4-AAP) and 3, 5-dichloro-2-hydroxybenzenesulfonate (DCHBS) under peroxidase catalysis to produce colored products. The system monitored the change in absorbance at 520nm at fixed time intervals. The change in absorbance is proportional to the concentration of uric acid in the sample.

## Potential confounding factors

We consider the following possible confounders associated with periodontitis and uric acid. Sociodemographic variables included age, sex, body mass index (BMI), race/ethnicity, level of education, and the ratio of household income to poverty, with education categorized into three categories: less than high school, high school, and bachelor's degree or higher. BMI was divided into obese (BMI$\geq$30kg/m2) and non-obese (BMI < 30kg/m2). The ratio of household income to poverty was divided into low (< 1.35), medium ($\geq$1.35, < 3.0) and high ($\geq$3.0). Lifestyle variables included smokers (never, current, quit).Alcohol intake can be divided into no (0 g/day), moderate alcohol consumption (0.1–27.9 g/day for men and 0.1–13.9 g/day for women) and heavy drinking ($\geq$28 g/day for men and $\geq$14 g/day for women).Drinking frequency was divided into no, sometimes (< once a week), regular ($\geq$ once a week).Frequency of floss use (never (0 number of uses), rarely (1–2 number of uses), moderate (3–5 number of uses), and frequent (6–7 number of uses). Physcian-diagnosed systemic diseases (hypertension, sleep disorders, high cholesterol), and diabetes mellitus (physcian-diagnosed, measured glycosylated hemoglobin >6.5%, measured fasting blood glucose $\geq$7mmol/L, and 2-hour glucose $\geq$11.1mmol/L after glucose tolerance test (OGTT)); Hyperlipidemia (measured LDL$\geq$130mg/dl and TC$\geq$240mg/dl).The intake of purine foods (seafood, pork, etc.) was divided into intake within 30 days and non-intake within 30 days.

## Statistical analysis

Statistical analyses were performed with the use of EmpowerStatsversion 4.1. Sample weights were calculated taking into account all NHANES estimates. Categorical variables were expressed as frequencies or percentages and compared using a weighted chi-square test, including sex, race, education, household income to poverty ratio, smoking history, drinking

history, flossing frequency, obesity, hypertension, diabetes, high cholesterol, hyperlipidemia, and sleep disorders. Continuous variables were mean ± SD and included age, sUA.

After adjusting for potential confounders, weighted multiple logistic regression model and smooth curve fitting were used to evaluate the association between sUA and moderate/severe periodontitis. Serum uric acid was categorized according to the quartile distribution of serum uric acid concentration (Q1:1.1–4.3 mg/dL; Q2:4.4–5.2 mg/dL; Q3:5.3–6.2 mg/dL; Q4:6.3–13.3 mg/dL), where the first quartile was designated as the reference group. Three statistical models were constructed: model 1, without covariate adjustment; Model 2, adjusted only for age and gender; Model III, adjusted for all covariates: age, sex, race, education, household income to poverty ratio, smoking history, drinking history, dental floss frequency, obesity, hypertension, diabetes, high cholesterol, hyperlipidemia, sleep disturbance, and finally, we examined gender and racial/ethnic subgroups to see if there were any sources of heterogeneity.

The weighted logistic regression model was used to calculate the difference of continuous variables. For categorical variables, a weighted chi-square test was used. A P value of less than 0.05 was considered statistically significant.

## Results

### Characteristics of the participants

There were no significant differences in education level and household income/poverty level ratio among the sUA groups (all p > 0.05).The mean age of the participants was 52.35 years, and 1865 (54.89%) were men and 1533 (45.11%) were women.Moderate or severe periodontitis was present in 42.5% of the participants. Participants in the Q4 group (individuals with high sUA concentrations) were more likely to be male, non-Hispanic white. Compared with the other groups, participants in Q4 group had higher education level, more alcohol consumption, lower frequency of dental floss, higher household income, higher proportion of poverty, and higher prevalence of obesity and cholesterol, hypertension, diabetes and hyperlipidemia (Table 1).

### Multiple regression analysis

The association between sUA and periodontitis is shown in Table 2. sUA as a continuous variable increased the prevalence of moderate/severe periodontitis, OR = 1.11, (CI: 1.06, 1.17), P<0.0001. Similar results were found after adjusting for age group and sex in Model II. In Model III, after adjusting for all covariates: age, sex, race, education, family income to poverty ratio, smoking history, alcohol intake and frequency, dental floss frequency, obesity, hypertension, diabetes, high cholesterol, hyperlipidemia, purine food intake and sleep disorders, sUA increased the prevalence of moderate to severe periodontitis with an OR of 1.10 (95%CI: 1.01, 1.21), P = 0.0287. This suggests that there may be a linear relationship between sUA and the prevalence of periodontitis.

To ensure the reliability of the results, sUA was associated with periodontitis in all models (Table 2). After transforming sUA from a continuous variable to a categorical variable (quartile), using the lowest quartile Q1 as a reference, the OR results of model II and III were consistent with the OR trend of unadjusted model 1. However, the P value of Q2 and Q3 groups was greater than 0.05, and Q4 group was less than 0.05. These results suggest that high level of serum uric acid is more related to moderate/severe periodontitis than low level of serum uric acid.

Subgroup analyses were performed to identify possible sources of heterogeneity (Fig 2). The interaction analysis showed that sUA was correlated with moderate/severe periodontitis in both men and women (OR = 1.083 and 1.148, P < 0.05). And sUA and non-hispanic whites

**Table 1. Weighted characteristics of the study population based on quartiles of serum uric acid.**

| Serum uric acid categories(%) | Q1(1.1–4.3) | Q2(4.4–5.2) | Q3(5.3–6.2) | Q4(6.3–13.3) | P-value |
|---|---|---|---|---|---|
| N | 777 | 866 | 852 | 903 | |
| Serum uric acid (mg/dL, mean ± SD) | 3.69 ± 0.52 | 4.82 ± 0.26 | 5.73 ± 0.29 | 7.24 ± 0.86 | <0.001 |
| Age (mean (SD)) | 50.33 ± 14.08 | 51.92 ± 15.06 | 53.18 ± 14.97 | 53.69 ± 15.09 | <0.001 |
| Sex (%) | | | | | <0.001 |
| Male | 372 (47.88%) | 445 (51.39%) | 499 (58.57%) | 549 (60.80%) | |
| Female | 405 (52.12%) | 421 (48.61%) | 353 (41.43%) | 354 (39.20%) | |
| Race (%) | | | | | <0.001 |
| Mexican-American | 129 (16.60%) | 123 (14.20%) | 120 (14.08%) | 89 (9.86%) | |
| Other Hispanics | 90 (11.58%) | 81 (9.35%) | 65 (7.63%) | 67 (7.42%) | |
| Non-Hispanic white | 318 (40.93%) | 390 (45.03%) | 391 (45.89%) | 388 (42.97%) | |
| Non-Hispanic black | 123 (15.83%) | 145 (16.74%) | 159 (18.66%) | 221 (24.47%) | |
| Other races | 117 (15.06%) | 127 (14.67%) | 117 (13.73%) | 138 (15.28%) | |
| Educational level (%) | | | | | 0.278 |
| Less than high school | 154 (19.82%) | 157 (18.13%) | 184 (21.60%) | 160 (17.72%) | |
| High school | 161 (20.72%) | 202 (23.33%) | 181 (21.24%) | 217 (24.03%) | |
| College graduate or above | 462 (59.46%) | 507 (58.55%) | 487 (57.16%) | 526 (58.25%) | |
| BMI categories(%) | | | | | <0.001 |
| Non-obese(<30 kg/m2) | 559 (72.60%) | 530 (61.41%) | 508 (59.91%) | 452 (50.11%) | |
| Obesity (> = 30 kg/m2) | 211 (27.40%) | 333 (38.59%) | 340 (40.09%) | 450 (49.89%) | |
| Household income to poverty ratio(%) | | | | | 0.151 |
| Low(<1.35) | 249 (35.27%) | 234 (29.51%) | 250 (31.45%) | 248 (29.35%) | |
| Middle(≥1.35,<3) | 177 (25.07%) | 211 (26.61%) | 194 (24.40%) | 232 (27.46%) | |
| High(≥3) | 280 (39.66%) | 348 (43.88%) | 351 (44.15%) | 365 (43.20%) | |
| Alcohol intake, (%) | | | | | <0.001 |
| None | 531 (73.96%) | 592 (73.82%) | 535 (67.30%) | 562 (66.59%) | |
| Moderate | 83 (11.56%) | 97 (12.09%) | 148 (18.62%) | 145 (17.18%) | |
| Not recorded | 104 (14.48%) | 113 (14.09%) | 112 (14.09%) | 137 (16.23%) | |
| Drink alcohol, (%) | | | | | <0.001 |
| No | 531 (74.58%) | 592 (74.75%) | 535 (67.81%) | 562 (67.39%) | |
| Sometimes (< once a week) | 57 (8.01%) | 63 (7.95%) | 80 (10.14%) | 62 (7.43%) | |
| Regularly (≥ once a week) | 124 (17.42%) | 137 (17.30%) | 174 (22.05%) | 210 (25.18%) | |
| Hypertension, n (%) | | | | | <0.001 |
| Yes | 214 (27.54%) | 301 (34.76%) | 336 (39.44%) | 460 (50.94%) | |
| No | 563 (72.46%) | 565 (65.24%) | 516 (60.56%) | 443 (49.06%) | |
| High cholesterol levels,n(%) | | | | | <0.001 |
| Yes | 212 (27.28%) | 339 (39.15%) | 354 (41.55%) | 424 (46.95%) | |
| No | 563 (72.46%) | 520 (60.05%) | 493 (57.86%) | 475 (52.60%) | |
| Not recorded | 2 (0.26%) | 7 (0.81%) | 5 (0.59%) | 4 (0.44%) | |
| Diabetes, n (%) | | | | | <0.001 |
| Yes | 677 (87.13%) | 721 (83.26%) | 705 (82.75%) | 714 (79.07%) | |
| No | 100 (12.87%) | 145 (16.74%) | 147 (17.25%) | 189 (20.93%) | |
| Hyperlipidemia,n(%) | | | | | <0.001 |
| Yes | 640 (82.37%) | 680 (78.52%) | 653 (76.64%) | 664 (73.53%) | |
| No | 137 (17.63%) | 186 (21.48%) | 199 (23.36%) | 239 (26.47%) | |
| Smoker(%) | | | | | <0.001 |
| Never smoker | 473 (60.88%) | 504 (58.20%) | 482 (56.57%) | 474 (52.49%) | |
| Active smoker | 156 (20.08%) | 164 (18.94%) | 158 (18.54%) | 143 (15.84%) | |

(*Continued*)

**Table 1.** (Continued)

| Serum uric acid categories(%) | Q1(1.1–4.3) | Q2(4.4–5.2) | Q3(5.3–6.2) | Q4(6.3–13.3) | P-value |
|---|---|---|---|---|---|
| Former smoker | 148 (19.05%) | 198 (22.86%) | 212 (24.88%) | 286 (31.67%) | |
| Sleep disorder, n (%) | | | | | <0.001 |
| Yes | 49 (6.31%) | 103 (11.89%) | 85 (9.98%) | 127 (14.06%) | |
| No | 728 (93.69%) | 763 (88.11%) | 767 (90.02%) | 776 (85.94%) | |
| Floss usage(%) | | | | | 0.010 |
| Never(0) | 206 (26.51%) | 237 (27.37%) | 245 (28.76%) | 312 (34.55%) | |
| Less use(1–2) | 126 (16.22%) | 137 (15.82%) | 133 (15.61%) | 153 (16.94%) | |
| Medium use(3–5) | 152 (19.56%) | 162 (18.71%) | 171 (20.07%) | 155 (17.17%) | |
| Frequent use(6–7) | 293 (37.71%) | 330 (38.11%) | 303 (35.56%) | 283 (31.34%) | |
| PERIODONTITIS | | | | | <0.001 |
| No/mild | 480 (61.78%) | 509 (58.78%) | 497 (58.33%) | 468 (51.83%) | |
| Moderate/severe | 297 (38.22%) | 357 (41.22%) | 355 (41.67%) | 435 (48.17%) | |
| Purine food intake | | | | | 0.108 |
| No (Within thirty days) | 203 (26.13%) | 222 (25.64%) | 186 (21.83%) | 206 (22.81%) | |
| Yes (Within thirty days) | 574 (73.87%) | 644 (74.36%) | 666 (78.17%) | 697 (77.19%) | |

Mean ±SD: P values for continuous variables were calculated from weighted logistic regression models. For categorical variables: P values were calculated by weighted chi-square tests.

and blacks in the correlation between moderate/severe periodontitis, OR 1.228 and 1.212, respectively, P values were significant ($P < 0.05$).

## Linear relationship exploration

After adjusting for potential confounders, curve-fitting analysis showed a linear relationship between sUA and periodontitis (Fig 3). As can be seen from the figure, the line generally shows an upward trend.

In the subgroup analysis, we found significant differences in the association between serum uric acid concentration and periodontitis by race/ethnicity and gender, which were further

**Table 2. Association between serum uric acid (mg/dL) and periodontitis.**

| Exposure | Adjust I β (95% CI) P value | Adjust IIβ (95% CI) P value | Adjust IIIβ (95% CI) P value |
|---|---|---|---|
| Serum uric acid (mg/dL, mean ± SD) | 1.11 (1.16, 1.17) <0.0001 | 1.10 (1.04, 1.16) 0.0003 | 1.10 (1.01, 1.21) 0.0287 |
| Serum uric acid categories(%) | | | |
| Q1(1.1–4.3) | 1.0 | 1.0 | 1.0 |
| Q2(4.4–5.2) | 1.13 (0.93, 1.38) 0.2149 | 1.13 (0.92, 1.38) 0.2551 | 1.17 (0.94, 1.53) 0.1354 |
| Q3(5.3–6.2) | 1.15 (0.95, 1.41) 0.1567 | 1.11 (0.90, 1.37) 0.3205 | 1.17 (0.92, 1.49) 0.2059 |
| Q4(6.3–13.3) | 1.50 (1.24, 1.83) <0.0001 | 1.44 (1.17, 1.76) 0.0005 | 1.49 (1.07, 2.08) 0.0193 |
| p-value for trend | <0.0001 | 0.0009 | 0.0395 |

Model 1: Unadjusted model.

Model 2: Adjusted for age, sex, and race/ethnicity.

Model 3: Adjusted for age, sex, race/ethnicity, education, household income/poverty ratio, smoking behavior, alcohol intake and frequency, dental floss frequency, obesity, hypertension, diabetes, high cholesterol, hyperlipidemia, purine food intake and sleep disorders.

| Sex | N | | OR | 95%CI | p–value |
|---|---|---|---|---|---|
| Male | 1865 | | 1.083 | (1.014, 1.156) | 0.01704 |
| Female | 1533 | | 1.148 | (1.066, 1.235) | 0.00024 |
| **Race** | | | | | |
| Mexican–American | 461 | | 1.093 | (0.951, 1.256) | 0.21044 |
| Other Hispanics | 303 | | 1.228 | (1.035, 1.458) | 0.01857 |
| Non–Hispanic white | 1487 | | 1.072 | (0.993, 1.158) | 0.07623 |
| Non–Hispanic black | 648 | | 1.204 | (1.078, 1.344) | 0.00098 |
| Other races | 499 | | 1.082 | (0.955, 1.226) | 0.21356 |

**Fig 2. Subgroup analyses of the association between moderate/severe periodontitis and quartiles of serum uric acid concentration by sex and race/ethnicity (NHANES 2009–2014; n = 3398).**

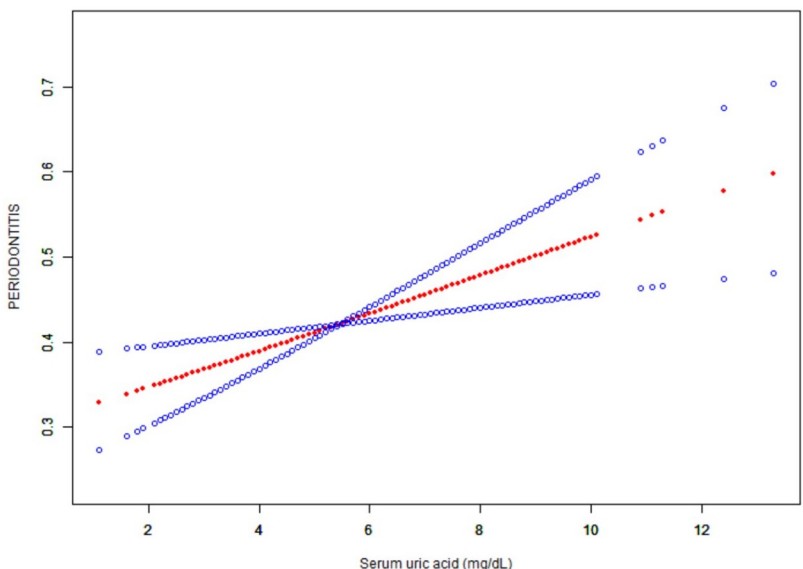

**Fig 3. A linear association was found between serum uric acid concentration and periodontitis.** Solid lines represent smooth curve fits between variables. The blue bands indicate 95% of the fitted confidence intervals. Adjustment for age, sex, race/ethnicity, education, household income/poverty ratio, smoking behavior, alcohol intake and frequency, dental floss frequency, obesity, hypertension, diabetes, high cholesterol, hyperlipidemia, purine food intake and sleep disorders.

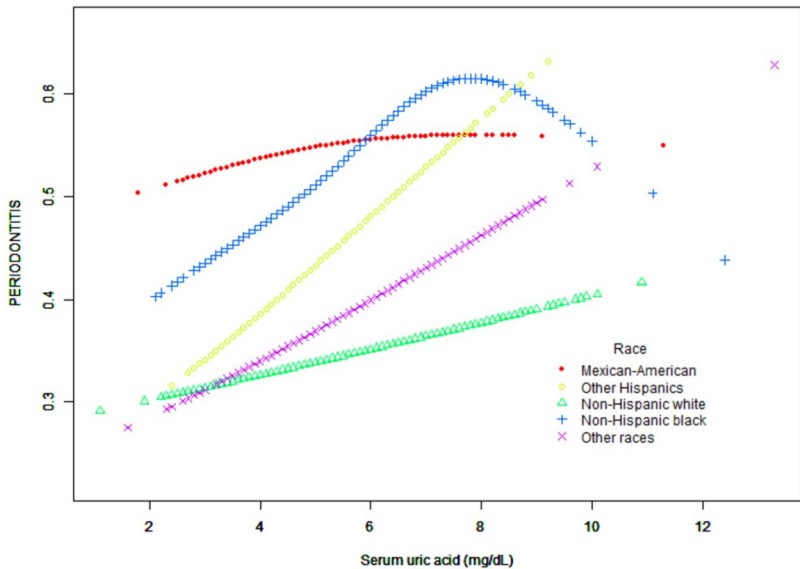

**Fig 4. Associations between uric acid and periodontitis stratified by race/ethnicity.**

analyzed using generalized additive model and smooth curve fitting (Figs 4 and 5). Curve-fitting analysis showed a linear relationship between sUA and periodontitis in other Hispanics, non-Hispanic whites, and other races, with a slow upward trend in Mexican Americans, while in non-Hispanic blacks, the association between serum uric acid and periodontitis was an inverted U-shaped curve, and a two-segment linear regression model was used to identify inflamation points, turning points: sUA 8.1mg/dL, $P < 0.05$. Unlike in men, the relationship between sUA and the prevalence of periodontitis in women followed a U-shaped curve with turning points of 4.3 mg/dL and 7.7mg/dL.

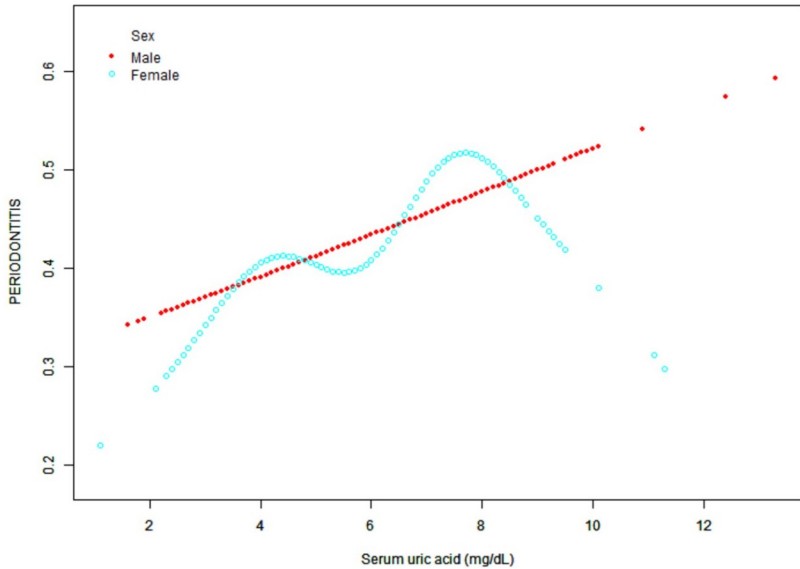

**Fig 5. The association between uric acid and periodontitis was stratified by sex.**

## Discussion

The current cross-sectional survey of a nationally representative sample of US participants was designed to investigate the association of sUA with periodontitis. After adjusting for all confounding factors, the results showed that there was a linear relationship between sUA and the prevalence of periodontitis, and as sUA levels increased, the prevalence of periodontitis also increased. This suggests that higher baseline sUA level is associated with the prevalence of moderate to severe periodontitis. In addition to this, we found several noteworthy points.

The first point was that, unlike in men, the association between sUA and the prevalence of periodontitis in women followed a U-shaped curve with tipping points of 4.3 mg/dL and 7.7 mg/dL determined by a two-stage linear regression model. Within this range, the prevalence of periodontitis decreased with increasing sUA concentration. This suggests that there may be a different relationship between sUA and the prevalence of periodontitis in different gender populations.

There is potential evidence that gender is a factor worth considering in the association between sUA and the prevalence of periodontitis. Sex steroids play a role in the prevalence of periodontitis [28]. Firstly, it is reflected in its effect on bone metabolism. Estrogen inhibits the expression of inflammatory cytokines that are important in bone resorption, and estrogen deficiency may lead to or block the progression of periodontitis [29–31]. Studies have shown that men with lower levels of bioavailable testosterone affected by sex hormone-binding globulin have a higher risk of periodontitis [28, 32]. The second is reflected in the function of sex steroid hormones regulating cells involved in the immune response. Females appear to have higher reactive and protective cell-mediated and humoral immune responses to antigenic challenge compared to males [33]. Women with periodontitis had higher levels of antibodies to Porphyromonas gingivalis than men [34]. This can be attributed to the attenuation of neutrophils in the presence of estrogen [35], as well as the inhibition of T cell and B cell lymphopoiesis and activation of B cell function by estrogen [36].

At the same time, the higher prevalence of chronic periodontitis in men reflects poor lifestyle or environmental factors, such as oral hygiene habits, higher cigarette smoking and lower use of oral health services [37]. However, the lack of sex-specific results in randomized controlled trials limits the understanding of sex-specific results after interventions related to periodontitis [38]. Therefore, in order to understand the mechanism behind the sex difference between sUA and periodontitis, further studies on the role of sex hormones are needed. Oral health care providers must be made aware of the gender bias contained in the evidence to take this into account in clinical decision making.

The second point is that the relationship between sUA and the prevalence of periodontitis in non-Hispanic blacks follows an inverted U-shaped curve with an inflection point of: 6.6mg/dL. Some studies have proposed that periodontitis patients with different racial/ethnic backgrounds have different oral microbial profiles [39]. Experiments have shown that black people have a stronger systemic neutrophil response to dental plaque accumulation [40]. Moreover, the difference in diet between non-Hispanic blacks and whites makes the reduction of serum uric acid level less effective than that of whites [41]. However, this is not consistent with our experimental results. To date, there is limited evidence linking sUA and periodontitis in different ethnic groups. We cannot explain this U-shaped curve from the existing evidence, and future prospective studies with large samples in different ethnic groups are needed to further verify it.

The third finding is that as can be seen in Table 1, the proportion of obese people was relatively low in Q1 and Q2 groups with lower uric acid levels, while the proportion of obese people increased in Q4 group with higher uric acid levels. Previously, Richard J. Johnson found that fructose-induced uric acid production in mice caused mitochondrial oxidative stress,

which stimulated fat accumulation [42]. Experiments in Ana Andres-Hernando mice found that, like fructose, uric acid accelerates the induction of obesity when ingested in excess of umami foods such as monosodium glutamate [43]. Therefore, people with higher uric acid levels can pay more attention to their diet to prevent or control obesity.

The relationship between UA and periodontitis is increasingly known [10, 19]. The current consensus is that pathological elevation of UA levels represents proinflammatory, oxidative, and osteoclast states [15]. Research reports in high uric acid hematic disease observed in children with elevated levels of proinflammatory cytokines [21]. Cabău G proposed that UA could induce epigenetic reprogramming and immune metabolism of innate immune cells to enhance inflammation, a process called training immunity [44]. UA can react with peroxynitrite to form free radicals [45]. It can also enhance intracellular superoxide production by increasing the activity of nicotinamide adenine dinucleotide phosphate (NADPH) oxidase [46]. This intracellular oxidative stress together with UA-induced inflammatory cytokines stimulates osteoclastic bone resorption and inhibits osteoblastic bone formation [15]. Experiments have shown that increased serum UA may reduce the pH value in the oral cavity or periodontal pocket, promote the growth and reproduction of acid-producing periodontal pathogens such as Prevotella, and increase the prevalence of periodontitis periodontitis [47]. Therefore, sUA clearly affects systemic inflammation, leading to chronic inflammatory disease, may increase the prevalence of periodontitis and supports our findings on the association between periodontitis and uric acid.

Compared with other similar studies, the advantage of this study is that it not only discussed the different concentration of sUA baseline characteristics of participants, and conducted a sUA concentrations and the linear regression analysis between the prevalence of periodontitis. Second, the sample size was large, including 3398 participants, including a representative sample of a multiethnic population, performing further subgroup analyses for sensitivity testing, and adjusting for many potential confounders, providing strong evidence for quantitative assessment of the association between sUA concentration and periodontitis index levels.

However, this experiment has some limitations. First, the limited amount of data available in the NHANES database prevented us from being more thorough about the effect of sUA on periodontitis. Second, because our results are based on the population of the United States, they may not apply to other countries. Third, this study was cross-sectional, and the causal relationship between sUA and periodontitis could not be clarified. Finally, bias due to other potential confounders that were not adjusted for in this study cannot be excluded.

## Conclusion

In conclusion, sUA levels were associated with moderate and severe periodontitis. However, the association between sUA levels and the occurrence of periodontitis in women and non-Hispanic blacks followed a U-shaped curve. sUA may directly or indirectly contribute to the global burden of periodontal disease, but there is little evidence for a directed association between sUA and periodontitis. Our study further confirmed that high sUA levels may be associated with moderately severe periodontitis, and if it can be used as evaluation of periodontitis risk or progress indicator provides a new way of thinking. More studies are needed to confirm the relationship between sUA and periodontitis.

## Supporting information

**S1 File. Data.**
(XLS)

## Author Contributions

**Conceptualization:** Jingjing Bai.

**Data curation:** Jingjing Bai.

**Formal analysis:** Jingjing Bai.

**Investigation:** Ming Ding, Zhonghua Zhang.

**Methodology:** Ming Ding, Zhonghua Zhang.

**Software:** Chenying Zhou, Ye Liu.

**Writing – original draft:** Jingjing Bai.

**Writing – review & editing:** Zhu Chen, Ping Feng, Jukun Song.

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
