## [Decision Letter · Decision Letter 0]

28 May 2024

PONE-D-24-03627Relationship between serum uric acid levels and periodontitisPLOS ONE

Dear Dr. Feng,

Thank you for submitting your manuscript to PLOS ONE. After careful consideration, we feel that it has merit but does not fully meet PLOS ONE’s publication criteria as it currently stands. Therefore, we invite you to submit a revised version of the manuscript that addresses the points raised during the review process.

"Besides the considerations of the reviewers, authors should pay attention to the appropriate terminology for a cross-sectional study, especially regarding its temporality."

We look forward to receiving your revised manuscript.

Kind regards,

Fernando Oliveira Costa, Ph.D.

Academic Editor

PLOS ONE

Journal Requirements:

https://www.hindawi.com/journals/cdtp/2021/4626062/

https://journals.sagepub.com/doi/10.1177/0022034509341013

https://coek.info/pdf-hyperuricemia-as-a-potential-plausible-risk-factor-for-periodontitis-.html

In your revision ensure you cite all your sources (including your own works), and quote or rephrase any duplicated text outside the methods section. Further consideration is dependent on these concerns being addressed.

4. PLOS requires an ORCID iD for the corresponding author in Editorial Manager on papers submitted after December 6th, 2016. Please ensure that you have an ORCID iD and that it is validated in Editorial Manager. To do this, go to ‘Update my Information’ (in the upper left-hand corner of the main menu), and click on the Fetch/Validate link next to the ORCID field. This will take you to the ORCID site and allow you to create a new iD or authenticate a pre-existing iD in Editorial Manager. Please see the following video for instructions on linking an ORCID iD to your Editorial Manager account: https://www.youtube.com/watch?v=_xcclfuvtxQ".

5. We note that your Data Availability Statement is currently as follows: [All relevant data are within the manuscript and its Supporting Information files]

6. Please remove your figures from within your manuscript file, leaving only the individual TIFF/EPS image files, uploaded separately. These will be automatically included in the reviewers’ PDF.

Additional Editor Comments:

Besides the considerations of the reviewers, authors should pay attention to the appropriate terminology for a cross-sectional study, especially regarding its temporality."

Reviewers' comments:

Reviewer's Responses to Questions

**Comments to the Author**

1. Is the manuscript technically sound, and do the data support the conclusions?

Reviewer #1: Partly

Reviewer #2: Partly

2. Has the statistical analysis been performed appropriately and rigorously? 

Reviewer #1: Yes

Reviewer #2: No

3. Have the authors made all data underlying the findings in their manuscript fully available?

Reviewer #1: Yes

Reviewer #2: Yes

4. Is the manuscript presented in an intelligible fashion and written in standard English?

Reviewer #1: Yes

Reviewer #2: No

5. Review Comments to the Author

Reviewer #1: The present project presents a relevant topic and a plausible justification. The introduction of the article provides scientific background and justification for the study, the methodological design is appropriate for the proposed objective, and the discussion contrasts the results with available references. However, it presents some limitations. Here are some considerations:

- Indicate the study design in the title.

- In lines 75 and 76, page 4, the authors cite reference 17 but state "studies have found". I suggest including all references to which the sentence refers.

- The use of diuretics and consumption of purine-rich foods, factors associated with serum uric acid levels, were not evaluated.

- Excessive alcohol consumption is associated with hyperuricemia. The assessment of alcohol consumption should be more rigorous and consider more categories, paying attention to frequency, quantity, and duration of consumption.

- Individuals with mild periodontitis are included in the comparison group along with individuals without periodontitis.

- In the introduction, the plausibility of the impact of serum uric acid levels on periodontitis is presented. In this sense, periodontitis should be the grouping variable.

- The impact of age on serum uric acid levels should be addressed in the discussion.

- A significant difference in relation to body mass index was observed among the four groups considering the quartile distribution of serum uric acid concentration. However, it was not reported between which groups the difference occurred. In addition, there seems to be no difference between obese and non-obese individuals in the Q4 group, with higher serum uric acid levels.

- Similarly, a significant difference in relation to periodontitis was observed among the four groups considering the quartile distribution of serum uric acid concentration. Report if there is a difference between no/mild and moderate/severe periodontitis in the Q4 group, with higher serum uric acid levels.

- In lines 206, page 15, the authors state that "the other three groups of sUA and periodontitis were associated with OR values of 1.17, 1.16, and 1.45, respectively." However, there is no significant difference for Q2 and Q3 in the three models presented.

- In Figure 2, the variable "non-Hispanic white" appears twice with different data.

- The authors state that it is a cross-sectional study, address the incidence of periodontitis, and present odds ratio (OR) data. In addition, they conclude that high uric acid levels are a risk factor for periodontitis.

Reviewer #2: Dear authors:

I read with great interest the article entitled Relationship between serum uric acid levels and periodontitis. The topic is relevant to periodontics and the health sector.

After a careful analysis, I concluded that the article in its current form is not suitable for publication and requires a detailed review by the authors on many items, which need to be clarified for better understanding by the reader. The study design is cross-sectional, and many terms used and statements are not suitable for a study with this type of design. Furthermore, details need to be incorporated into the text for important items and to improve understanding of the study.

Therefore, I am attaching the PDF file of the article highlighted in yellow and comments on some items that I consider important and are not clear throughout the text. Do this to improve the article.

6. PLOS authors have the option to publish the peer review history of their article (what does this mean?). If published, this will include your full peer review and any attached files.

Reviewer #1: No

Reviewer #2: No

---

## [Author Response · Author response to Decision Letter 0]

16 Aug 2024

Rebuttal letter to the Academic Editor

1、Please ensure that your manuscript meets PLOS ONE's style requirements, including those for file naming. 

Response: Thanks for your advice! I have modified the format as required.

2、We noticed you have some minor occurrence of overlapping text with the following previous publication(s), which needs to be addressed.

Response: Thank the editor for giving me the suggestion, I have finished the revision. It is your professional questions that make my article more complete.

3、We note that the grant information you provided in the ‘Funding Information’ and ‘Financial Disclosure’ sections do not match.

Response: Thank you for your question. I have modified it when I submitted it again.

4、ORCID iD 

Response:I have linked the ORCID iD to the Editor Manager account as requested.

5、minimal data set

Response: Thank you for reminding me, I couldn't agree more. For the minimal data set in my article (including my graphs, tables, and data), I have uploaded the Supporting information file.

6、Please remove your figures.

Response: I have removed the figures from the manuscript as requested and uploaded them as files, named "S1_Fig", "S2_Fig", "S3_Fig", "S4_Fig" and "S5_Fig".

7、Authors should pay attention to the appropriate terminology for a cross-sectional study, especially regarding its temporality.

Response: Thank you for your professional advice. My article is a cross-sectional study, and I acknowledge that this type of study can only demonstrate an association between serum uric acid and periodontitis, without establishing a causal relationship. I have revised the manuscript to address any ambiguities related to this issue.

The hypothesis proposed in my manuscript, that "uric acid is a risk factor for periodontitis," is grounded in previous research:

Evidence 1: Uric acid has its own mechanism of enhancing inflammation.

Evidence 2: Elevated uric acid levels may reduce the pH value in the oral cavity or periodontal pocket, promoting the growth and reproduction of acid-producing periodontal pathogens such as Prevotella, thereby increasing the prevalence of periodontitis.

Evidence 3: Some studies have observed improvement in periodontitis following the control of uric acid levels.

(More details are available in the manuscript.)

Rebuttal letter to Reviewer 1:

1.Indicate the study design in the title.

Response: Thank you for your professional suggestion. I have indicated in the title that this is a cross-sectional study. This will make my research more clear to the reader.

2.In lines 75 and 76, page 4, the authors cite reference 17 but state "studies have found". I suggest including all references to which the sentence refers.

Response: Thank you for your professional suggestions. First, I revised the original phrase "elevated serum UA level or changes in sialopurine metabolism in periodontitis patients" to "uric acid may play a role in the inflammatory pathology of periodontitis," which better conveys my intended meaning.

Secondly, I realized that using only a single document is insufficient to support my point. In response, I have added relevant studies. However, including all relevant literature would be overwhelming and cumbersome. Therefore, I have included only a few representative and persuasive studies:

1.Increased Uric Acid Levels in CP Patients: Studies have shown that plasma uric acid levels are significantly increased in CP patients compared to healthy individuals. (Correlation of Toll-like receptor 4, interleukin-18, transaminases, and uric acid in patients with chronic periodontitis and healthy adults)

2.Mouse Experiment: In a mouse experiment, PD induction caused more severe bone destruction in hyperuricemic mice compared to normouricemic mice. (Obesity-Related Gut Microbiota Aggravates Alveolar Bone Destruction in Experimental Periodontitis through Elevation of Uric Acid)

3.Hyperuricemia and Periodontitis: The effect of hyperuricemia on periodontitis is as follows: periodontitis is an inflammatory disease caused by periodontal pathogen infection, while hyperuricemia is a metabolic disease caused by excessive soluble UA in the blood, leading to aseptic inflammation. (Recent progress and perspectives on the relationship between hyperuricemia and periodontitis).

3.The use of diuretics and consumption of purine-rich foods, factors associated with serum uric acid levels, were not evaluated.

Response: Thank you for your professional proposal. After your suggestion, I actively searched for relevant information in the database. However, as this survey is based on the conditions available at that time, it may not be entirely comprehensive regarding the investigation content.

First, I was unable to find relevant surveys on the use of diuretics in the database, so I regret to inform you that I cannot incorporate this aspect into the study. I hope you understand.

Secondly, there is also no clear investigation or data on the consumption of purine-rich foods. However, I did find some relevant data, such as the intake of foods high in purines (e.g., seafood, pork, etc.). Unfortunately, data on the consumption of all purine-rich foods were not included.

Based on the available data, I analyzed the relationship between the intake of the mentioned foods and uric acid levels by categorizing them into two groups: intake within 30 days and non-intake within 30 days. The results were not significant, but this may be due to the limitations of not including all purine-rich foods in the study.

4.Excessive alcohol consumption is associated with hyperuricemia. The assessment of alcohol consumption should be more rigorous and consider more categories, paying attention to frequency, quantity, and duration of consumption.

Response: Thank you for your professional suggestion, excessive drinking is indeed related to hyperuricemia. In response to your suggestion, I would like to subdivide alcohol consumption into none (0 g/day), moderate (0.1-27.9 g/day for men and 0.1-13.9 g/day for women), and heavy (≥28 g/day for men and ≥14 g/day for women). The frequency of drinking was divided into never drinking, sometimes drinking (< 1 time/week) and often drinking (≥1 time/week). According to univariate analysis, the results showed that alcohol intake and frequency were associated with serum uric acid.

5.Individuals with mild periodontitis are included in the comparison group along with individuals without periodontitis.

Response: Response: Periodontitis is a dichotomous variable, and I have divided it into two groups: no/mild periodontitis and moderate/severe periodontitis. This categorization method is based on recommendations from the Centers for Disease Control and Prevention (Recent epidemiologic trends in periodontitis in the USA). Similar approaches have been used in most comparable studies.

For example:

Association between Dietary Pattern and Periodontitis -- A Cross-Sectional Study.

Does Periodontitis Affect the Association of Biological Aging with Mortality?

Mean Platelet Volume is associated with periodontitis: A cross-sectional study.

The association of composite dietary antioxidant index with periodontitis in NHANES 2009-2014.

6.In the introduction, the plausibility of the impact of serum uric acid levels on periodontitis is presented. In this sense, periodontitis should be the grouping variable.

Response: Thank you for your careful review of my research. In my study, periodontitis was defined as a dichotomous variable: none/mild periodontitis and moderate/severe periodontitis. I have made revisions regarding the ambiguities in the expression of my manuscript.

7.The impact of age on serum uric acid levels should be addressed in the discussion.

Response: Thank you for your professional advice. As for the effect of age on serum uric acid level, I searched the literature and found that the direct or indirect evidence that age has an effect on serum uric acid level is very limited. There is even an article showing that Serum uric acid does not increase with age (Serum uric acid in relation to age and physique in health and in coronary heart disease). This leads to insufficient evidence to support my discussion. I don't know whether I misunderstood or did not find the relevant study you want. I look forward to hearing from you if you have any suggestions.

8.A significant difference in relation to body mass index was observed among the four groups considering the quartile distribution of serum uric acid concentration. However, it was not reported between which groups the difference occurred. In addition, there seems to be no difference between obese and non-obese individuals in the Q4 group, with higher serum uric acid levels.

Response: Thank you for your serious proposal. In Table I, in Q1 and Q2 groups with lower uric acid levels, the proportion of obese people was relatively low, while in Q4 group with higher uric acid levels, the proportion of obese people increased. By searching the literature, there is a significant relationship between uric acid and obesity, so it brings us enlightenment: people with higher uric acid levels can pay more attention to their own obesity control.

9.Similarly, a significant difference in relation to periodontitis was observed among the four groups considering the quartile distribution of serum uric acid concentration. Report if there is a difference between no/mild and moderate/severe periodontitis in the Q4 group, with higher serum uric acid levels.

Response: Thank you for your attention to this question. Regarding the univariate analysis, it was used to look for confounding factors that might have had an impact on the experiment. The association between no/mild periodontitis and serum uric acid levels was the focus of my entire study, so I excluded it from my univariate analysis. In the later multiple regression analysis, I adjusted for other factors that may have influenced the experiment, such as: age, sex, race/ethnicity, education, household income/poverty ratio, smoking behavior, alcohol intake and frequency, floss frequency, obesity, hypertension, diabetes, high cholesterol, hyperlipidemia, purine food intake, and sleep disorders. The results showed that when serum uric acid was a continuous variable, it was associated with periodontitis. When serum uric acid was used as a categorical variable, it could be seen that there was an association between the two in the Q4 group with high serum uric acid levels. And I did a linear analysis, which showed a linear relationship. The above analysis shows that there is a certain relationship between serum uric acid and moderate to severe periodontitis.

10.In lines 206, page 15, the authors state that "the other three groups of sUA and periodontitis were associated with OR values of 1.17, 1.16, and 1.45, respectively." However, there is no significant difference for Q2 and Q3 in the three models presented.

Response: Thank you for your question, I fully understand your question. What I want to express here is the overall trend that compared with Q1 group, the risk of periodontitis increased with the increase of serum uric acid level. However, I may not have expressed myself clearly and let people have a misunderstanding. So I changed my wording to: “After transforming sUA from a continuous variable to a categorical variable (quartile), using the lowest quartile Q1 as a reference, the OR results of model II and III were consistent with the OR trend of unadjusted model I. However, the P value of Q2 and Q3 groups was greater than 0.05, and Q4 group was less than 0.05. These results suggest that high level of serum uric acid is more related to moderate/severe periodontitis than low level of serum uric acid.”

11.In Figure 2, the variable "non-Hispanic white" appears twice with different data.

Response: Thank you for your careful reminder. I made a mistake when I typed. I can't imagine how much of a problem it would be without your reminding! I've changed it to "Other Hispanics."

12.The authors state that it is a cross-sectional study, address the incidence of periodontitis, and present odds ratio (OR) data. In addition, they conclude that high uric acid levels are a risk factor for periodontitis.

Response: Thank you for your professional question. In this cross-sectional study, serum uric acid was found to be associated with moderate/severe periodontitis. And according to previous studies, serum uric acid can increase the risk of periodontitis, so I mentioned that serum uric acid is a risk factor for periodontitis. This is really not rigorous, and as I mentioned the study limitations, I cannot know the causal relationship between serum uric acid and periodontitis. Therefore, the article has been corrected to read: serum uric acid is associated with moderate/severe periodontitis.

Rebuttal letter to Reviewer 2:

1. For all wording changes.

Response: Thank you for your advice, with your advice, my article can be more complete and careful. As for my inappropriate wording, I also revised it. 

1. "unadjusted covariates" was replaced by "unadjusted model"

2. I changed "but there is little evidence for a directed association between sUA and periodontitis" in my article to "but there is little evidence that sUA is directly related to periodontitis”

3."correlation" was changed to "association".

2. The numbers in my Figure 1 don't match

Response: You are a respectable and meticulous professor. I am truly sorry for my carelessness and have made the necessary corrections immediately. Thanks to your keen observation, I was able to avoid such a significant mistake.

3.The definition of moderate to severe periodontitis met the CDC/AAP definition (2017).

Response: Thank you for your suggestion. By reviewing relevant literature and similar articles. I will change this "Moderate periodontitis was defined as 2 interproximal sites with AL ≥ 4 mm (not on the same tooth), or ≥ 2 interproximal sites with PD ≥ 5 mm (not on the same tooth). Severe periodontitis was defined as 2 interproximal sites with AL ≥ 6 mm (not on the same tooth) and ≥ 1 interproximal site with PD ≥ 5 mm”

4.Citing the assessment method of uric acid concentration in this study.

Response: I'm basing this on data from the NHANES database, which is publicly available. I have added the evaluation method as per your requirements:

Centers for Disease Control and Prevention. National Health and Nutrition Examination Survey, 2009-2010 Data Documentation, Codebook, and Frequencies, Standard Biochemistry Profile (BIOPRO_F). Accessed 13 May 2022. Available from https://wwwn.cdc.gov/nchs/nhanes/2009-2010/BIOPRO_F.htm.

5.How can the authors make this claim just by presenting association measurement findings? For this purpose, linear regression analysis is necessary. Please justify this statement.

Response: Thank you for your question. Initially, I used multiple regression analysis and adjusted the model, which resulted in OR=0.11, (95% CI: 1.03, 1.16), P =0.002. However, I recognize that this alone does not sufficiently support the conclusion that the risk of moderate-to-severe periodontitis increases by 10% for each unit increase in serum uric acid level.

To further validate this claim, I conducted a linear regression analysis, which revealed a positive linear correlation between serum uric acid levels and moderate-to-severe periodontitis. This finding supports my previous conclusion. However, I agree that it is inappropriate to make a definitive conclusion based solely on the association measurements presented earlier. Therefore, I have revised the statement to: "This suggests that there may be a linear relationship between serum uric acid levels and the prevalence of periodontitis."

6.The authors' statements are not confirmed by the findings in Table 2. Please justify.

Response: Thank you for your question. In Table II, when serum uric acid was changed into a categorical variable. In model 3, when Q1 group was used as the reference, the OR of Q2 group, Q3 group, and Q4 group were 0.17, 0.17, and 0.49, respectively (variables were added according to the opinions of another reviewer). Only the Q4 group was statistically significant. It can only be said that the overall trend of Q2, Q3, and Q4 groups was consistent, and the high level of serum 

---

## [Decision Letter · Decision Letter 1]

28 Aug 2024

Relationship between serum uric acid levels and periodontitis —A Cross-Sectional Study

PONE-D-24-03627R1

Dear Dr. Feng,

We’re pleased to inform you that your manuscript has been judged scientifically suitable for publication and will be formally accepted for publication once it meets all outstanding technical requirements.

An invoice will be generated when your article is formally accepted. Please note, if your institution has a publishing partnership with PLOS and your article meets the relevant criteria, all or part of your publication costs will be covered. Please make sure your user information is up-to-date by logging into Editorial Manager at Editorial Manager and clicking the ‘Update My Information' link at the top of the page. If you have any questions relating to publication charges, please contact our Author Billing department directly at authorbilling@plos.org.

Kind regards,

Fernando Oliveira Costa, Ph.D.

Academic Editor

PLOS ONE

Additional Editor Comments (optional):

The suggestions were accepted and the paper was accepted for publication in PlosONeBest regards

Reviewers' comments:

Reviewer's Responses to Questions

**Comments to the Author**

1. If the authors have adequately addressed your comments raised in a previous round of review and you feel that this manuscript is now acceptable for publication, you may indicate that here to bypass the “Comments to the Author” section, enter your conflict of interest statement in the “Confidential to Editor” section, and submit your "Accept" recommendation.

Reviewer #1: All comments have been addressed

2. Is the manuscript technically sound, and do the data support the conclusions?

Reviewer #1: Yes

3. Has the statistical analysis been performed appropriately and rigorously? 

Reviewer #1: Yes

4. Have the authors made all data underlying the findings in their manuscript fully available?

Reviewer #1: Yes

5. Is the manuscript presented in an intelligible fashion and written in standard English?

Reviewer #1: Yes

6. Review Comments to the Author

Reviewer #1: (No Response)

7. PLOS authors have the option to publish the peer review history of their article (what does this mean?). If published, this will include your full peer review and any attached files.

Reviewer #1: No

---

## [Editor Report · Acceptance letter]

18 Sep 2024

PONE-D-24-03627R1 

PLOS ONE

Dear Dr. Feng, 

I'm pleased to inform you that your manuscript has been deemed suitable for publication in PLOS ONE. Congratulations! Your manuscript is now being handed over to our production team.

Kind regards, 

on behalf of

Professor Fernando Oliveira Costa 

Academic Editor

PLOS ONE